# NLRP3 Inflammasome as a Potential Therapeutic Target in Dry Eye Disease

**DOI:** 10.3390/ijms241310866

**Published:** 2023-06-29

**Authors:** Dian Zhuang, Stuti L. Misra, Odunayo O. Mugisho, Ilva D. Rupenthal, Jennifer P. Craig

**Affiliations:** 1Department of Ophthalmology, New Zealand National Eye Centre, Faculty of Medical and Health Sciences, The University of Auckland, Auckland 1142, New Zealand; dian.zhuang@auckland.ac.nz (D.Z.); s.misra@auckland.ac.nz (S.L.M.); 2Buchanan Ocular Therapeutics Unit, Department of Ophthalmology, New Zealand National Eye Centre, Faculty of Medical and Health Sciences, The University of Auckland, Auckland 1142, New Zealand; lola.mugisho@auckland.ac.nz (O.O.M.); i.rupenthal@auckland.ac.nz (I.D.R.)

**Keywords:** dry eye disease, NLRP3 inflammasome, inflammation, ocular surface, tear film, anti-inflammatory treatment

## Abstract

Dry eye disease (DED) is a multifactorial ocular surface disorder arising from numerous interrelated underlying pathologies that trigger a self-perpetuating cycle of instability, hyperosmolarity, and ocular surface damage. Associated ocular discomfort and visual disturbance contribute negatively to quality of life. Ocular surface inflammation has been increasingly recognised as playing a key role in the pathophysiology of chronic DED. Current readily available anti-inflammatory agents successfully relieve symptoms, but often without addressing the underlying pathophysiological mechanism. The NOD-like receptor protein-3 (NLRP3) inflammasome pathway has recently been implicated as a key driver of ocular surface inflammation, as reported in pre-clinical and clinical studies of DED. This review discusses the intimate relationship between DED and inflammation, highlights the involvement of the inflammasome in the development of DED, describes existing anti-inflammatory therapies and their limitations, and evaluates the potential of the inflammasome in the context of the existing anti-inflammatory therapeutic landscape as a therapeutic target for effective treatment of the disease.

## 1. Introduction

Dry eye disease (DED) is a growing global chronic ocular surface disorder [1] with two main subtypes: aqueous deficient dry eye (ADDE), where lacrimal gland dysfunction presents most severely in autoimmune disorders such as Sjögren syndrome (SSDE) [2], and evaporative dry eye (EDE), where meibomian gland dysfunction (MGD) is considered the leading cause in clinical practice and population-based studies [3,4,5]. Dysfunction of either the lacrimal or meibomian glands leads to a compromised tear film and therefore damage to the ocular surface [6]. More recently, decreased wettability dry eye has been proposed by the Asia Dry Eye Society as the third subtype of DED [7]. Due to a deficiency of membrane-associated mucin, ocular surface wettability is reduced, contributing to reduced tear film stability, which may shorten the tear film breakup time. As a multifactorial disease, any aetiology of DED will have one or multiple entry points into a self-propagating pathway that perpetuates the disease state.

The concept of the DED vicious circle recognises inflammation as both a cause and consequence of the disease [8]. Anti-inflammatory agents are used to disrupt the vicious circle in an attempt to restore homeostasis of the tear film and ocular surface and reduce the signs and symptoms of DED [9,10]. Current agents include topical medications such as corticosteroids and cyclosporin A, oral antibiotics such as tetracycline and macrolides, essential fatty acids (EFAs), and autologous serum. However, a number of these agents are associated with a risk of side effects that increase with extended use, including ocular irritation [10,11].

An emerging body of evidence supports the role of the NOD-like receptor protein-3 (NLRP3) inflammasome, a key driver in the innate immune system, in DED pathogenesis [12,13,14]. Recent studies have explored the changes in inflammasome-related activities in patients with DED and murine models compared to controls, as well as the suppression of the NLRP3 inflammasome pathway. This review discusses currently available anti-inflammatory therapies for DED, highlights the involvement of the inflammasome in the development of DED, and evaluates its potential to serve as a therapeutic target for DED.

## 2. The Vicious Circle of DED

Under normal conditions, transient ocular surface dryness is well defended via the lacrimal functional unit [8], an integrated system that maintains ocular surface homeostasis. Chronic gland dysfunctions can cause sustained surface desiccation and tear film instability, leading to tear hyperosmolarity, a recognised hallmark of DED [15]. Hyperosmolar tears can damage the ocular surface by initiating a cascade of inflammatory signalling events in epithelial cells. Inflammatory mediators and proteases are released, driving epithelial and goblet cell loss and glycocalyx disruption, observed clinically as punctate epitheliopathy and tear film instability [16]. This premature tear film breakup, in turn, promotes further tear hyperosmolarity and amplifies ocular surface damage, thus perpetuating the pathological cycle widely recognised as the vicious circle of DED (Figure 1) [2,15,17,18]. Ultimately, entry into the circle drives patient reports of ocular discomfort and poor quality of vision secondary to the loss of tear film homeostasis [1,15,17].

## 3. Inflammation in Dry Eye Disease

Over the past 15 years, a growing body of literature has established a key role for inflammation in the pathophysiology of DED, as evidenced in tissue culture, animal models, and patients with DED [19]. Ocular-surface-related inflammation was first identified in the lacrimal glands of patients with Sjögren syndrome (SSDE), where lymphocytic infiltration of the exocrine glands (including the lacrimal glands) leads to replaced or destroyed secretory acini, and hence reduced tear secretion [20]. Later evidence shows ageing as a common risk factor for inducing an inflammatory response in the lacrimal glands in non-Sjögren syndrome dry eye (NSSDE) [16]. It is now understood that inflammation is not only present in the lacrimal glands but also in the conjunctiva in patients with DED, with and without Sjögren syndrome [21]. DED-related inflammation involves both innate and adaptive immune responses. The initial innate immune response on the ocular surface occurs when a trigger such as environmental stress disturbs the dynamics of the tear film and the associated lacrimal glands, neural network, and ocular surface. This, in turn, activates a signalling cascade mediated by mitogen-activated protein kinases (MAPK) in ocular surface epithelial cells [22,23]. Transcription factors such as nuclear factor kappa B (NF-κB), activator protein 1 (AP-1), and activating transcription factor (ATF) are then stimulated, initiating the release of pro-inflammatory cytokines, chemokines, and matrix metalloproteinases (MMPs) [16]. In response to the inflammatory environment on the ocular surface, immature antigen-presenting cells (APCs) are subsequently activated, transitioning to mature cells, and these migrate to regional lymphoid tissues to initiate the adaptive immune response [24]. Mature APCs subsequently initiate the adaptive immune response by priming naïve T cells in the lymphoid compartments, leading to the expansion of autoreactive CD4+ helper T cell subsets, Th1 and Th17 [24,25]. These effector cells then migrate to and infiltrate the ocular surface to release additional pro-inflammatory mediators that facilitate lymphocytic infiltration, promoting further ocular surface damage (Figure 2) [22,26].

Cytokines are key signalling molecules that mediate intercellular communication in response to a stimulus such as injury, while chemokines are cytokines that regulate the directed migration of immune cells via chemotaxis. A number of pro-inflammatory cytokines and chemokines have been implicated in the pathogenesis of inflammatory DED. Interleukin (IL)-1β, IL-6, IL-8, tumour necrosis factor alpha (TNF)-α, and transforming growth factor beta (TGF)-β are pro-inflammatory cytokines that regulate immune and inflammatory processes on the corneal epithelium via c-Jun N-terminal kinases (JNK) and extracellular-regulated kinase (ERK) MAPK signalling pathways [21,23,28,29,30,31,32]. Interferon-gamma (IFN-γ), the hallmark cytokine of Th1 cells, is induced via MAPK and NF-κB under hyperosmolar conditions. Increased IFN-γ exacerbates DED-induced conjunctival cell apoptosis via both the caspase-8-mediated extrinsic pathway and the caspase-9-mediated intrinsic pathway following desiccating stress [33]. The differentiation of IL-17 is induced by IL-6, IL-23, and TGF-β, which promote the production of MMPs by corneal epithelial cells and contribute to corneal epithelial barrier dysfunction [25]. IL-10 is an anti-inflammatory cytokine that suppresses the activation of T cells [34]. Studies have shown significantly upregulated levels of cytokines, including IL-1β, IL-6, IL-8, IL-10, IL-17, IL-18, IFN-γ, TNF-α, and TGF-β in experimental DED models [35,36,37,38,39,40] as well as in the tears and conjunctival tissues of DED patients [41,42,43,44,45,46]. In addition, the concentrations of cytokines studied showed a positive correlation with disease severity as established either according to the TFOS DEWS criteria or by symptoms only, based on the OSDI scales [43,47].

MMPs are endopeptidases that are involved in physiological extracellular matrix remodelling and, pathologically, in wound healing, angiogenesis, and cancer [48]. Tear hyperosmolarity stimulates the production of MMP-1, -3, -9, and -13 from corneal epithelial cells through the activation of the JNK signalling pathway [49]. Moreover, MMP-9 modulates DED severity by disrupting the tight junctions in the corneal epithelium, initiating T cell recruitment, and stimulating the release of additional inflammatory cytokines, contributing to the perpetuation of the DED vicious circle [9,50]. Studies reported raised levels of pro-MMP-9 and MMP-9 in the tears of patients with DED [45,47,51]. Interestingly, MMP inhibitors, TIMP-1 and TIMP-2, have been shown to mediate corneal epithelial proliferation and inhibit apoptosis in a rabbit DED model [52].

Human leukocyte antigen D-related (HLA-DR) is a major histocompatibility complex (MHC) class II cell surface receptor involved in antigen presentation via binding to intracellularly processed peptides and presenting them to T cells [53]. It is used as a reliable inflammatory biomarker for immune activation and disease severity in ocular surface disorders [54]. IFN-γ upregulates HLA-DR expression in conjunctival cells, leading to increased antigen presentation and activation of T cells and hence a downstream inflammatory state in conjunctival cells [55,56]. Elevated HLA-DR expression by conjunctival epithelial cells has been detected in patients with moderate-to-severe DED [57,58,59,60].

Cell adhesion molecules (CAMs) are cell-surface proteins that facilitate cell migration by binding to components within the extracellular matrix [61]. Intercellular adhesion molecule-1 (ICAM-1) is known to play a role in the migration of lymphocytes into lymph nodes or inflamed tissues such as the ocular surface of DED patients [62]. Significantly higher levels of ICAM-1 are expressed by the conjunctival epithelium in DED, which strongly correlates with HLA-DR expression [21,58].

## 4. The Inflammasome Pathway

The innate immune system relies on pattern-recognition receptors (PRRs) to recognise invading pathogens. A group of PRRs, named the nucleotide-binding oligomerization-(NOD)-like receptors (NLRs), are recognised to form part of the innate immune response [63]. The inflammasome is an intracellular protein complex in which NLRs are important sensor proteins. The NLR family members are characterised by the presence of a tripartite structure: a central NOD, a pyrin domain (PYD) at the amino terminus, and multiple leucine-rich repeats (LRRs) at the carboxyl terminus [63]. NLRP3 is the best characterised of the 14 members of the inflammasome family and is an important regulator of inflammatory diseases [64]. Apart from the sensor protein, the NLRP3 inflammasome also consists of an adaptor protein, an apoptosis speck-like protein (ASC), and an effector protein (pro-caspase-1) (Figure 3) [65].

The NLRP3 inflammasome is assembled following a two-step process that involves priming and activation [66]. A priming signal can be pathogen-associated molecular patterns (PAMPs) such as bacteria, danger-associated molecular patterns (DAMPs) such as heat shock proteins, adenosine triphosphate (ATP), or a range of external stimuli such as environmental stress [67,68]. In the development of DED, hyperosmolarity, as a priming signal for the inflammatory cascade, is detected by the innate immunocytes via PRRs [39,49]. The priming process activates NF-κB and subsequently upregulates the expression of inflammasome-related components, including inactive NLRP3, ASC, pro-caspase-1, pro-forms of IL-1β, and IL-18 [69]. Following the priming step, the second step involves a response to several upstream stimuli to induce full activation and inflammasome formation. These stimuli include extracellular ATP, potassium efflux, calcium signalling, and reactive oxygen species (ROS) production. Upon activation, pro-caspase-1 undergoes autolytic cleavage to active caspase-1, which in turn cleaves pro-IL-1β and pro-IL-18 into their bioactive pro-inflammatory forms that regulate downstream immune responses (Figure 4) [70,71]. Caspase-1 also cleaves the pore-forming protein, gasdermin D (GSDMD), forming pores on the plasma membrane, promoting membrane rupture, and finally leading to pyroptosis, a form of programmed cell death [72,73,74].

Growing evidence has shown the importance of NLRP3 in inflammatory and autoimmune diseases, playing a key role in the innate immune system [75,76]. The NLRP3 has been implicated in various ocular diseases, including age-related macular degeneration [77,78], diabetic retinopathy [79,80], Behçet’s disease [81], uveitis [82,83], and ocular hypertension [84]. Over the past decade, the NLRP3 inflammasome has been increasingly implicated in the pathogenesis of DED [12,13,85]. Risk factors such as ageing, desiccating environments, and contact lens wear contribute to inflammasome activation. ROS has been suggested as a priming signal for inflammasome activation with environmental induction [12]. Hyperosmolar stress on the ocular surface induces ROS production, which has been found to be elevated in corneal epithelial cells in animal models and in tear and conjunctival samples of patients with DED [86,87,88]. Elevation in ROS generation is a proximal signal for NLRP3 inflammasome activation, subsequently resulting in the release of the proinflammatory cytokine IL-1β [89]. An overview of studies that have examined associations between inflammasome activation and DED is presented in Table 1.

The contribution of the ROS-NLRP3-IL-1β signalling pathway to the pathogenesis of environment-induced DED has previously been demonstrated both in vivo and in vitro [12,13]. Exposure to a hyperosmotic environment induces ROS generation, which acts as a priming signal to activate the downstream NLRP3-caspase-1 signalling pathway. The activated pathway mediates innate immune responses and subsequently results in increased IL-1β secretion [12]. IL-1β functions as a key pro-inflammatory cytokine mediator in the innate and adaptive immune systems in DED and is believed to act as an early indicator of DED onset [93]. IL-1β also induces the differentiation and promotes maintenance of Th17 cells, which are the main effector cells that cause epithelial damage in DED [94]. Similar findings were observed in a clinical study in which mRNA and protein levels of NLRP3, caspase 1, IL-1β, and IL-18 were elevated in tears and conjunctival samples from patients with DED, in particular SSDE, compared to controls [14]. Another study revealed a novel innate immune pathway where, in addition to the well-known NLRP3 activation, the anti-inflammatory NLRP6 pathway was suppressed, contributing to the inflammatory response in DED models [85]. The imbalanced activation of NLRP3/NLRP6 inflammasomes in the surface epithelium provides insight into the innate immune mechanism of the ocular surface in response to environmental stress [85]. Similar findings were shown in a study using human corneoscleral tissue, where inhibition of transient receptor potential melastatin 2 (TRPM2), an osmotic stress biosensor, suppressed the hyperosmolar-induced increases of NLRP3, caspase-1, and IL-1β expression [90].

## 5. Current Anti-Inflammatory Therapies for DED

A wide range of management strategies for DED exist, such as topical eye drops [95]. With inflammation having a recognised role in the pathophysiology of chronic DED, numerous anti-inflammatory agents have been investigated as a means of disrupting the vicious circle of DED and attempting to restore ocular surface homeostasis [9,10]. Topical anti-inflammatory agents such as corticosteroids, cyclosporin A, oral tetracyclines, omega-3 and -6 fatty acids, and monoclonal antibodies, understood to target specific factors within the inflammatory cascade, are further discussed.

Topical corticosteroids, such as methylprednisolone and fluorometholone, are potent immunosuppressors that reduce pro-inflammatory cytokine and chemokine activation while enhancing anti-inflammatory gene expression through either the traditional glucocorticoid receptor-mediated pathways or non-receptor pathways [96,97,98,99]. Clinical studies have identified the efficacy of topical corticosteroid use in both effecting significant symptom relief and clinical sign remission, including diminishing corneal fluorescein staining and improving fluorescein tear clearance following a treatment course of two to three weeks [100,101,102]. Extended use of topical corticosteroids is, however, associated with an increased risk of serious side effects such as increased intraocular pressure (IOP) and development of cataracts and glaucoma, as well as an increased risk of microbial infection [10]. Topical corticosteroids that penetrate the corneal tissue less effectively, such as fluorometholone, have shown effectiveness in the reduction of ocular surface signs and symptoms with a lower tendency to induce IOP elevation compared with steroids such as methylprednisolone [103,104]. A synthetic topical fluorometholone nanoformulation offers superior therapeutic efficacy with one-fifth of the dosage of a commercial fluorometholone preparation (FML^®^, Allergan, an AbbVie company, Madison, NJ, USA) while reducing drug-related side effects [105]. Eysuvis^®^ (Kala Pharmaceuticals, Arlington, MA, USA), an ophthalmic suspension containing 0.25% loteprednol etabonate, was approved by the US Food and Drug Administration (FDA) for short-term management of DED in 2020 [106]. Multicentre randomised clinical studies showed better penetration and absorption at the ocular surface compared to the conventional 0.05% drug suspension. As a result, a lower concentration could be used while still achieving high efficacy and proving to be safe and well-tolerated for between two and four weeks of use [107,108]. In addition, a novel class of anti-inflammatory compounds, selective glucocorticoid receptor agonists (SEGRAs), regulate glucocorticoid receptor-mediated gene expression and have demonstrated anti-inflammatory properties with an improved side-effect profile relative to conventional corticosteroids [109].

Non-steroidal anti-inflammatory drugs (NSAIDs), including diclofenac, ketorolac, pranoprofen, and indomethacin, have been used topically for DED therapy [110,111,112]. Diclofenac has been shown to suppress hyperosmolarity-induced cell damage and alleviate ocular symptoms in DED patients [113,114]. However, due to the risk of delayed wound healing and re-epithelialization, impairment of corneal sensitivity, and risk of corneal melting, NSAIDs are generally not considered a viable option for the long-term treatment of DED and should not be prescribed without caution and close monitoring [3,110,114].

Cyclosporin A is an immunomodulator used most commonly in topical form to treat DED. Within the USA, an ophthalmic emulsion of 0.05% cyclosporin A was approved by the FDA in 2003, and the first generic cyclosporin ophthalmic eye drops were approved in 2022 [115]. Cyclosporin A binds to calcineurin and inhibits IL-2 transcription, preventing T cell activation and associated immune responses [116]. Its immunosuppressive effect was first demonstrated in the canine spontaneous DED model [117]. Numerous clinical trials have since been conducted to investigate the efficacy of topical cyclosporin A as a DED treatment for humans. Significant improvements have been demonstrated in patient-reported symptoms, tear break-up time, corneal fluorescein staining, Schirmer test results with anaesthesia suggesting improved lacrimal gland function, as well as frequency of artificial tear administration following long-term use of 0.05% or 0.1% cyclosporin A [118,119,120,121]. In addition, reduced expression of the immune activation marker HLA-DR and inflammatory cytokines such as IL-6 has been observed [122,123]. Drawbacks from prolonged use of the 0.05% cyclosporin A emulsion include the common side effects of ocular irritation and a burning sensation on instillation [11]. The release of free surfactants from the breakdown of the emulsion may also cause irritation to the ocular surface [124]. Several new commercialised formulations have been developed to address challenges with user tolerance and low bioavailability of cyclosporin A in the form of Restasis^®^ (Allergan, an AbbVie company, Madison, NJ, USA), including Ikervis^®^ (Santen Pharmaceuticals Co., Ltd., Osaka, Japan) and Vevye^®^ (Novaliq GmbH, Heidelberg, Germany) [125].

Tetracycline and its analogues, including doxycycline and minocycline, are oral antibiotics with both antibacterial and anti-inflammatory properties. Prescribed in low doses over an extended treatment period of around three months, tetracyclines have demonstrated the potential to manage severe MGD and break the vicious circle of DED by inhibiting inflammatory responses and counteracting free fatty acid accumulation in MGD development [3]. Clinical investigations of oral tetracyclines for chronic blepharitis and MGD have shown significantly improved patient-reported symptoms and ocular inflammatory signs [3,126,127], but not without the risk of significant side effects that include gastrointestinal symptoms and photosensitivity, tooth discoloration, and bacterial resistance to infection [128].

Macrolides are antibiotics that possess anti-inflammatory properties and can be used topically or orally to manage DED. Azithromycin inhibits zymosan-induced release of pro-inflammatory cytokines including TNF-α and IL-1β, chemokines including IL-8, and MMPs including MMP-1, -3, and -9 in human corneal epithelium by suppressing NF-κB activation [129]. Significant improvement in signs and symptoms of MGD and evaporative DED was observed between two and four weeks of topical azithromycin application [130,131]. Studies favour five-day oral azithromycin treatment over one-month oral doxycycline due to superior clinical improvement, fewer side effects, lower cost, and shorter duration of treatment in patients with MGD, but more research is warranted, particularly with respect to confirming the optimal dosage regime [132].

Tacrolimus, also known as FK506, is a macrolide antibiotic with potent immunosuppressive properties [133]. It possesses a similar mechanism of action to cyclosporin A but is between 10 and 100 times more potent [133,134]. Topical application of tacrolimus 0.03% every 12 h has demonstrated significant improvements in tear film stability and clinical signs of tear deficiency in SSDE patients [135].

The EFAs, omega-3 and -6, are essential for regulating ocular surface inflammation and homeostasis. Current Western diets typically feature high ratios of omega-6 to omega-3, often approaching a ratio of 17:1, promoting the pathogenesis of many diseases [136]. Omega-6 EFAs are the precursors of the pro-inflammatory eicosanoids prostaglandin E2 and leukotriene B4, whereas omega-3 EFAs produce largely anti-inflammatory prostaglandin E3, leukotriene B5, thromboxane, and resolvins [137]. Significant improvements in symptoms and signs, including Schirmer test score and tear break-up time, have been reported with three month administration of omega-3 supplementation, suggesting a positive effect on the lacrimal gland and tear film [138,139]. Reduced expression of HLA-DR was observed following three month supplementation of omega-3 and -6 [140]. However, more recent studies have reported no difference between omega-3 and placebo supplement groups on symptoms and signs of DED [141,142,143]. Finally, a recent Cochrane review concluded that there is a possible role for long-chain omega-3 supplementation in DED management, although with uncertainty and inconsistency in the evidence [144].

Lifitegrast was approved by the FDA in 2016 as a topical treatment for DED. It targets inflammation in DED by blocking the binding of the cognate ligand ICAM-1 to the integrin, leukocyte function-associated antigen 1 (LFA-1) [145]. Several preclinical studies and clinical trials have demonstrated suppression of cytokine release and significant improvement of signs and symptoms of DED with a favourable long-term safety profile. Irritation is commonly reported at the instillation site but is deemed to be within acceptable tolerance limits, and there have been no unexpected adverse events [145,146,147,148,149].

Tofacitinib, also known as CP-690550, is a potent selective inhibitor of the JAK family, which is critically involved in immune cell activation and pro-inflammatory cytokine production [150]. Clinical studies have demonstrated significant reductions in the immune activation marker HLA-DR and pro-inflammatory cytokines following eight weeks of topical tofacitinib treatment [151]. Improved signs and symptoms of DED were also reported after the treatment, along with a reasonable safety profile and tolerability [152].

Autologous serum, derived from the patients’ own blood serum, contains growth factors, bactericidal components, cytokine inhibitors, and anti-inflammatory cytokines in abundance and exhibits antibacterial and anti-inflammatory properties [153,154,155]. It is more effective than artificial tears in improving clinical signs and symptoms in patients with severe DED [153,156]. A recent Cochrane review concluded some benefit in improving patient-reported symptoms using 20% autologous serum in the short term; however, evidence of improvement over longer periods was lacking [157].

Herbal agents such as *Aster koraiensis* extract have been studied in dry eye models and have been shown to inhibit inflammation in the corneal epithelium and the lacrimal gland in animal models [158]. More research is therefore needed in clinical settings.

Thus, while anti-inflammatory medications show clinical and symptomatic benefits in DED, they often fall short of providing safe, long-term relief. Associated side effects can pose sight-threatening risks and discomfort to patients. Further research and development are thus needed to overcome these limitations and optimise the efficacy, tolerability, and safety of anti-inflammatory therapy for DED.

## 6. Inflammasome Inhibitors for DED

An increased understanding of the inflammasome pathways in ocular surface diseases through ongoing research has brought to light the potential for a novel, more targeted approach to DED treatment. Currently, studies are underway to investigate the possible therapeutic effects of various anti-inflammatory molecules on DED models via the NLRP3 inflammasome-mediated signalling pathway [12,159,160,161,162,163,164,165]. Table 2 summarises published pre-clinical studies investigating potential therapies that target the inflammasome pathway.

Various studies have noted that agents including NAC, calcitriol, polydatin, disulfiram, hADSC, SM934, combinations of CMC and α-MSH, linarin, and mADSC-exosomes reduced inflammasome-associated inflammation in several in vitro and in vivo DED models (Table 1). These agents target a range of mediators along the inflammasome pathway, such as NF-κB, caspase, ROS, IL-1β, and IL-18. Animal studies have shown that topical application of SM934 and mADSC-Exos suppresses the NLRP3 inflammasome pathway, thereby alleviating inflammation in DED models [162,165]. NAC significantly reduced ROS production in experimental mice and HCECs, alleviating ocular surface signs through NLRP3 inflammasome downregulation [12,13]. Later treatment with calcitriol showed comparable results to treatment with NAC in inhibiting hyperosmolar-induced oxidative stress both in vitro and in vivo, although some discrepancies in effect were observed, suggesting multifunctional properties of calcitriol [92,159]. Similar findings have also been reported using treatments with polydatin, combined CMC and α-MSH, hADSCs, and mADSC-exosomes, although further studies are warranted to elucidate the precise mechanisms as well as confirm safety [160,161]. Linarin was observed to alleviate corneal epithelial damage and suppress NLRP3 inflammasome-mediated immunity in a murine DED model without cytotoxicity, but its clinical efficacy relative to commercially available anti-inflammatory eye drops requires further investigation [164]. The stability and solubility of SM934 in aqueous solutions favour its applicability in DED management [162]. The combined treatment with α-MSH and CMC proved to be superior to treatment with conventional CMC alone, although other effects of this combination warrant further research [163]. More studies are needed to assess the impact of blocking the various mediators within the pathway.

## 7. Conclusions

DED is a complex disease that can be caused by a wide variety of factors. Evidence confirms the significant role of inflammation in DED development and progression. Anti-inflammatory agents such as corticosteroids and NSAIDs provide symptomatic relief in the short term, but their lack of specificity fails to address the underlying cause of DED while risking adverse effects from long-term use. Cyclosporin A is an effective immunomodulator and has proven safer than corticosteroids in the longer term; however, limitations in tolerability, a long onset of action, and low bioavailability are major drawbacks. The activation of inflammasome pathways, especially the ROS-NLRP3-IL-1β signalling axis, has been recognised as a novel but key mechanism involved in ocular surface inflammation, expanding the range of possible drug targets that might facilitate control of inflammation and reduce the burden of DED. However, the full extent of the role of the NLRP3 inflammasome activation in DED and its mechanisms of action remain the subject of ongoing investigation. More pre-clinical and clinical research is warranted to evaluate inflammasome inhibitors that might best serve as viable therapeutics for the treatment of DED.

## Figures and Tables

**Figure 1 ijms-24-10866-f001:**
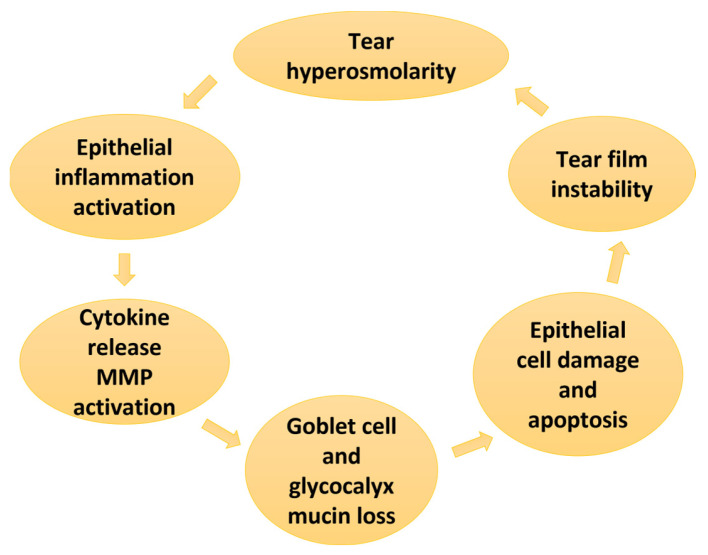
Key elements that feature in the vicious circle of DED as described in the TFOS DEWS II Pathophysiology report [2].

**Figure 2 ijms-24-10866-f002:**
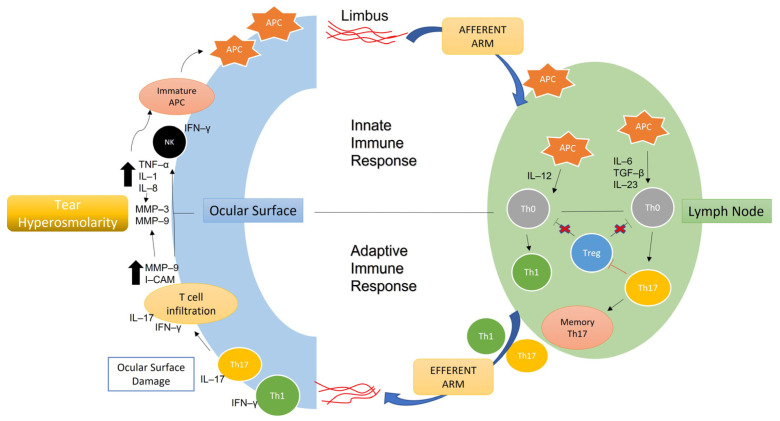
Inflammatory pathways in DED, adapted with permission from Tsubota et al. [27]. Copyright 2020, MDPI. Abbreviations: APC, antigen-presenting cells; IFN-γ, interferon gamma; IL, interleukin; I-CAM, intercellular adhesion molecule; MMP, matrix metalloproteinase; NK, natural killer cell; TGF-β, transforming growth factor beta; Th0, T helper cell type 0; Th1, T helper cell type 1; Th17, T helper cell type 17; Treg, regulatory T cell. Black arrows refer to upregulation of pro-inflammatory mediators; red crosses indicate the suppressive function of the regulatory T cells (Treg).

**Figure 3 ijms-24-10866-f003:**
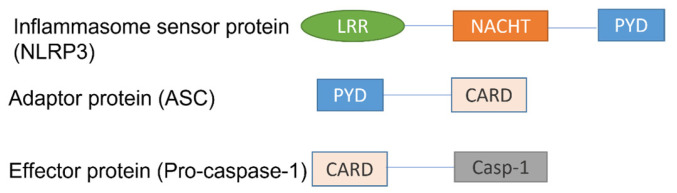
Structure of the NLRP3 inflammasome complex, consisting of a sensor protein (NLRP3), an adaptor protein (ASC), and an effector protein (pro-caspase-1). LRR, leucine-rich repeats; NACHT, a central nucleotide-binding and oligomerisation domain; PYD, a pyrin domain; and CARD, a caspase-recruitment domain.

**Figure 4 ijms-24-10866-f004:**
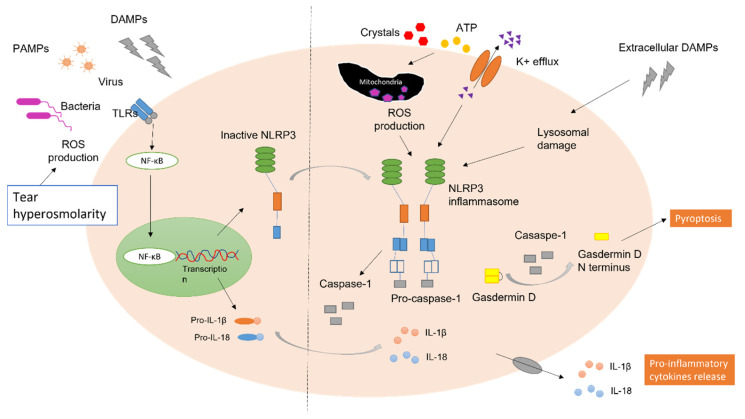
NLRP3 inflammasome activation pathway in response to tear hyperosmolarity, adapted from Groslambert and Py [66].

**Table 1 ijms-24-10866-t001:** Overview of studies investigating inflammasomes in DED.

Author	Title	DED Model	Findings
Pre-clinical studies			
Zheng et al. [12]	Reactive oxygen species activated NLRP3 inflammasomes prime environment-induced murine dry eye.	Murine model and HCECs.	Direct association between ROS generation and DED development; Increased ROS generation triggered NLRP3 inflammasome-mediated caspase-1 activation, resulting in elevated IL-1β expression; ROS inhibition alleviated DED symptoms through downregulation of NLRP3 inflammasomes and IL-1β.
Zheng et al. [13]	Reactive oxygen species-activated NLRP3 inflammasomes initiate inflammation in hyperosmolarity-stressed human corneal epithelial cells and environment-induced dry eye patients.	HCECs, tear, and conjunctival cells.	Elevated ROS production under hyperosmotic exposure; subsequent increased expression of the NLRP3 inflammasome as a ROS sensor, triggering the innate immune responses; pro-caspase-1, pro-IL-1β gene expression, and IL-1β secretion increased in tears and conjunctival cells of environment-induced DED patients.
Chi et al. [85]	Mitochondrial DNA oxidation induces the imbalanced activity of NLRP3/NLRP6 inflammasomes by activating caspase-8 and BRCC36 in the dry eye.	HCECs and murine models.	ROS-induced mitochondrial DNA oxidative damage induced imbalanced activation of NLRP3/NLRP6 inflammasomes (activated NLRP3 and suppressed NLRP6) via stimulation of caspase-8 and BRCC36 under hyperosmolar stress in the ocular surface epithelium of both cultured cells and animal models; resultant caspase-1 activation and IL-1β and IL-18 secretion led to ocular surface inflammation.
Zheng et al. [90]	Hyperosmotic stress-induced TRPM2 channel activation stimulates NLRP3 inflammasome activity in primary human corneal epithelial cells.	HCECs.	Hyperosmolar medium-induced activation of TRPM2, an osmotic and oxidative stress sensor; subsequently, increased NLRP3 inflammasome activity and IL-1β protein expression in HCECs.
Chen et al. [91]	NLRP12- and NLRC4-mediated corneal epithelial pyroptosis is driven by GSDMD cleavage accompanied by IL-33 processing in the dry eye.	Murine model and HCECs.	Presentation of GSDMD expression and induced IL-33 maturation in the corneal epithelium of the dry eye model; desiccating stress-induced activation of NLRP12/NLRC4 inflammasomes initiated GSDMD-dependent pyroptosis, driven by TLR4-mediated caspase-8 signalling.
Clinical studies			
Niu et al. [14]	Upregulation of the NLRP3 inflammasome in the tears and ocular surface of dry eye patients.	Tear and conjunctival samples of SSDE and NSSDE patients.	First study to investigate changes in NLRP3 inflammasome levels in tears and ocular surface cells of SSDE and NSSDE patients; upregulated levels of mRNA and protein expression of the NLRP3 inflammasome as well as downstream caspase-1, IL-1β, and IL-18 in tear samples and conjunctival impression cytology specimens of patients with DED, especially SSDE.
Zhang et al. [92]	Calcitriol alleviates hyperosmotic stress-induced corneal epithelial cell damage via inhibiting the NLRP3-ASC-caspase-1-GSDMD pyroptosis pathway in dry eye disease.	HCECs and tear samples from patients with DED and controls.	Elevation of GSDMD N-terminal domain expression showed the presentation of active pyroptosis in the tears of DED patients and hyperosmotic stress-induced HCECs. Inhibition of pyroptosis by disulfiram reduced the amount of HCECs with pyroptosis markers compared to those only exposed to the hyperosmotic medium.

Abbreviations: ASC, apoptosis speck-like protein; BRCC36, BRCA-1-BRCA-2-containing complex; DED, dry eye disease; GSDMD, gasdermin D; HCECs, human corneal epithelial cells; IL, interleukin; NLRC4, NLR family CARD domain-containing protein 4; NLRP3, NOD-like receptor protein-3; NSSDE, non-Sjögren syndrome dry eye; ROS, reactive oxygen species; SSDE, Sjögren syndrome dry eye; TLR4, toll-like receptor 4; TRPM2, transient receptor potential melastatin 2.

**Table 2 ijms-24-10866-t002:** Studies investigating possible inflammasome inhibitors to treat the inflammatory component of DED.

Author	Title	DED Model	Findings
Zheng et al. [12]	Reactive oxygen species activated NLRP3 inflammasomes prime environment-induced murine dry eye.	Murine model.	Application of eye drops containing 0.3% NAC markedly reduced ROS production and subsequently downregulated the expression of NLRP3 inflammasome components and IL-1β in mouse eyes compared to the control group under a desiccating environment.
Zheng et al. [13]	Reactive oxygen species-activated NLRP3 inflammasomes initiate inflammation in hyperosmolarity stressed human corneal epithelial cells and environment-induced dry eye patients.	HCECs, HCCs, and tears.	ROS inhibitor, NAC, suppressed the hyperosmolarity-induced ROS-NLRP3-IL-1β signalling pathway in HCECs, blocking rises in NLRP3 inflammasome formation, caspase-1 activity, and IL-1β release.
Dai et al. [159]	Calcitriol inhibits the ROS-NLRP3-IL-1β signalling axis via activation of Nrf2-antioxidant signalling in hyperosmotic stress stimulated human corneal epithelial cells.	HCECs.	Calcitriol, the active metabolite of vitamin D_3_, markedly suppressed the expression and activation of the NLRP3 inflammasome and the production of IL-1β in response to hyperosmotic stress; a protective effect of calcitriol observed against the DED-related inflammatory response via the NRF-2 antioxidant signalling.
Park et al. [160]	Polydatin inhibits the NLRP3 inflammasome in DED by attenuating oxidative stress and inhibiting the NF-κB pathway.	Murine model and HCCs.	Polydatin, a major active component in *Polygonum cuspidatum*, suppressed hyperosmolar stress-induced NLRP3 inflammasome activation by inhibiting NF-kB signalling pathways and ROS production; reduced mRNA expression levels of the NLRP3 inflammasome and pro-inflammatory cytokines, including MMP-9, IL-6, TNF-α, and IL-1β observed with polydatin treatment in vivo and in vitro.
Yu et al. [161]	hADSCs-derived extracellular vesicles inhibit NLRP3 inflammasome activation and dry eye.	Murine model and HCECs.	Reduced inflammatory cytokine production in topical hADSC-extracellular vesicle-treated mice was observed compared to vehicle control. A protective effect in HCECs noted by suppressing desiccating stress-induced NLRP3 inflammasome formation, caspase-1 activation and IL-1β maturation following hADSC-EVs treatment in vitro.
Zhang et al. [92]	Calcitriol alleviates hyperosmotic stress-induced corneal epithelial cell damage via inhibiting the NLRP3-ASC-caspase-1-GSDMD pyroptosis pathway in dry eye disease.	HCECs.	Calcitrol effectively alleviated hyperosmotic stress-induced corneal epithelial cell damage by inhibiting the NLRP3-ASC-caspase-1-GSDMD pyroptosis signalling pathway.
Yang et al. [162]	The artemisinin analogue SM934 alleviates dry eye disease in rodent models by regulating TLR4/NF-κB/NLRP3 signalling.	Murine model.	Topical application of β-aminoarteether maleate (SM934) (0.1% and 0.5%) significantly reduced levels of inflammatory cytokines (TNF-α, IL-1β, IL-6, IL-10) and MMP-9 in the ocular surface of SCOP- and BAC-induced rodent models. Possible association identified between the mechanism of action of SM934 and its direct inhibitory effect on the TLR4/NF-κB/NLRP3 inflammatory pathway in macrophages.
Lv et al. [163]	A combination of CMC and α-MSH inhibited the ROS-activated NLRP3 inflammasome in hyperosmolarity stressed HCECs and scopolamine-induced dry eye rats.	Murine model and HCECs.	Significant increase in protein levels of NLRP3 and caspase-1 was noted in the SCOP-induced-DED model; upregulated mRNA transcription of ROS, NLRP3, and IL-1β was observed in HCECs under hyperosmotic stimulation; downregulated ROS level, NLRP3 inflammasome, and caspase-1 were identified in vivo and in vitro under the combined treatment of CMC, a high-viscosity polymer, and MSH, an immunoregulatory neuropeptide.
Chen et al. [164]	Linarin ameliorates innate inflammatory response in an experimental DED model via modulation of the NLRP3 inflammasome.	Murine model and HCECs.	Inhibitory effect of topical linarin, a natural flavonoid glucoside with anti-inflammatory properties, on the NLRP3 inflammasome was observed by the reduced expression of NLRP3, ASC, caspase-1, IL-1β, and IL-18 in the conjunctiva of DED mice.
Wang et al. [165]	Exosomes derived from mouse adipose-derived mesenchymal stem cells alleviate the benzalkonium chloride-induced mouse dry eye model via inhibiting the NLRP3 inflammasome.	Murine model.	Treatment with mADSC-exosomes suppressed the NLRP3-associated inflammatory response by significantly reducing the gene and protein expression of NLRP3, caspase-1, IL-1β, and IL-18 in the conjunctiva of a BAC-induced mouse dry eye model.

Abbreviations: BAC, benzalkonium chloride; CMC, carboxylmethylcellulose; DED, dry eye disease; hADSC-EVs, human adipose tissue stem cells derived extracellular vesicles; HCCs, human conjunctival cells; HCECs, human corneal epithelial cells; IL, interleukin; mADSC-Exos, mouse adipose-derived mesenchymal stem cell-derived exosomes; MMP, matrix metalloproteinase; MSH, α-melanocyte stimulating hormone; NAC, N-acetyl-L-cysteine; NLRP3, NOD-like receptor protein-3; NRF2, nuclear factor erythroid 2-related factor 2; ROS, reactive oxygen species; SCOP, scopolamine hydrobromide; TLR4, toll-like receptor 4.

## Data Availability

No new data were created in the development of this manuscript.

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
