# Peer review of "NLRP3 Inflammasome as a Potential Therapeutic Target in Dry Eye Disease"

_ijms, 2023, doi:10.3390/ijms241310866_

Round 1

Reviewer 1 Report

Authors well-reviewed the underlying inflammatory mechanisms of dry eye disease (DED) and anti-inflammatory therapies, especially well-presented the inflammasome related pathways with high-quality figures and tables.

The potential NLRP3 inflammasome inhibitors for DED are summarized clearly and shown with detailed information, including the key findings, dosage, and the associated signaling pathways. 

Overall, it is a high-quality manuscript, providing promising insights that target to NLRP3 inflammasome for DED treatment.

The quality of English is fine. Few typos need to be corrected.

Author Response

Reviewer #1:

Authors well-reviewed the underlying inflammatory mechanisms of dry eye disease (DED) and anti-inflammatory therapies, especially well-presented the inflammasome related pathways with high-quality figures and tables.

The potential NLRP3 inflammasome inhibitors for DED are summarized clearly and shown with detailed information, including the key findings, dosage, and the associated signaling pathways.

Overall, it is a high-quality manuscript, providing promising insights that target to NLRP3 inflammasome for DED treatment.

Response: We thank the reviewer for their positive comments. We have corrected the typos in the manuscript, including updating “TFN” to “TNF” (line 120) and changing “dry eye disease” to “DED” in lines 42, 50, 70, 81, 101, 107, 113, 131, 134, 225, 234, 280, 359, 372, 380, 389, 393, 415, 425 and 427.

Reviewer 2 Report

Authors could discuss more about the traditional use of herbal agents to stimulate more production of exocrine glands in the GI tract and elsewhere like lachrymal glands and salivary glands.

Additional comments

(1) Authors focused on anti-inflammation molecular pathways, which are related but not solely causing DED

(2) DED should include discussions about the secretory function of the exocrine glands responsible.

(3) Authors might take reference to other publication targeting more general aspects of DED.

(4) Authors could have more general discussions

(5) References are solid

NA

Author Response

Reviewer #2:

Authors could discuss more about the traditional use of herbal agents to stimulate more production of exocrine glands in the GI tract and elsewhere like lachrymal glands and salivary glands.

  • Authors focused on anti-inflammation molecular pathways, which are related but not solely causing DED.
  • DED should include discussions about the secretory function of the exocrine glands responsible.
  • Authors might take reference to other publication targeting more general aspects of DED.
  • Authors could have more general discussions
  • References are solid

Response: We thank the reviewer for their suggestion regarding the inclusion of the secretory function of exocrine glands and traditional use of herbal agents for lacrimal gland stimulation in our manuscript. While we acknowledge the importance of the secretory function of exocrine glands in dry eye disease, our manuscript specifically focuses on the inflammasome involved in the disease pathogenesis, aiming to provide a comprehensive discussion of the immune-related pathways and pro-inflammatory mediators. We have therefore carefully considered your valuable insights and suggestions and updated the following inclusion:

  • “where lacrimal gland dysfunction” in the introduction (line 28)
  • “Dysfunctions to both the lacrimal and meibomian glands lead to compromised tear film and therefore damage to the ocular surface” (line 31-33)
  • “where lymphocytic infiltration of the exocrine glands (including lacrimal glands) leads to replaced or destroyed secretory acini, hence reduced tear secretion [20]. Later evidence shows aging as a common risk factor for inducing inflammatory response in lacrimal glands in non-Sjögren syndrome dry eye (NSSDE)” (line 76-80)
  • “suggesting improved lacrimal gland function” (line 284-285)
  • “, suggesting a positive effect on the lacrimal gland and tear film” (line 337-338)
  • “Herbal agents such as koraiensis extract has been studied in dry eye models and has been shown to inhibit inflammation in the corneal epithelium and the lacrimal gland in animal models [158]. More research is therefore needed in clinical settings.” (line 372-375)

Reviewer 3 Report

This is an excellent review that provides a comprehensive understanding on NLRP3 inflammasome regarding dry eye disease as therapeutic targets. The author extended the explanation on role of inflammation in dry eye disease. This review is instructive for both basic researchers and clinicians in this area.

There are some tiny minor concerns that needs to be addressed before publication.

Line 31. There is glowing evidence of the presence of other special subtype, called decreased wettability dry eye disease. This subtype of dry eye disease can be included briefly in addition to aqueous deficient type dry eye and evaporative dry eye in the introduction.

Line 37 etc. The term, "dry eye disease" has already been defined before this portion.

The abbreviation, "DED" should be used here. The abbreviation should be used consistently throughout the text.

Line 44 etc. “Dry eye “should be used as the abbreviation "DED" as defined by the authors. Many portions are found that “DED” should be used instead of dry eye or dry eye disease (Line 72, 98, 104, 122---, Table 1-) Please have a consistency and revise throughout the text.

Line 111.  “TFN” should be replaced with “TNF”.

After page 5, there are no line numbers.

The authors mentioned “NSAIDs are generally not considered viable option inf the long-term treatment of DED and should not be prescribed without caution and close monitoring [3,108,112].” What does the phrase,“viable option inf the long term treatment” mean? Please explain the meaning of a word, “inf”.

Reference No.113 The description of the reference No.113 is incomplete. Example: FDA, FDA, ----- sicca (., etc.

Reference No.141. The description of the reference No.141 is incomplete. (Author’s name)

Author Response

Reviewer #3:

Line 31. There is glowing evidence of the presence of other special subtype, called decreased wettability dry eye disease. This subtype of dry eye disease can be included briefly in addition to aqueous deficient type dry eye and evaporative dry eye in the introduction.

Response: Thank you for your comments. We have added ‘decreased wettability dry eye disease” in the introduction – “More recently, decreased wettability dry eye has been proposed by the Asia Dry Eye Society as third subtype of DED [6]. Due to deficiency of membrane-associated mucin, ocular surface wettability is reduced, contributing to reduced tear film stability which may shorten tear breakup time” (line 33-36).

Line 37 etc. The term, "dry eye disease" has already been defined before this portion.

The abbreviation, "DED" should be used here. The abbreviation should be used consistently throughout the text.

Line 44 etc. “Dry eye “should be used as the abbreviation "DED" as defined by the authors. Many portions are found that “DED” should be used instead of dry eye or dry eye disease (Line 72, 98, 104, 122---, Table 1-) Please have a consistency and revise throughout the text.

Response: We agree with reviewer’s comment. We have updated “dry eye disease” to “DED” throughout the paper (please see line 42, 50, 70, 81, 101, 107, 113, 131, 134, 225, 234, 280, 359, 372, 380, 389, 393, 415, 425 and 427).

Line 111.  “TFN” should be replaced with “TNF”.

Response: Thank you for spotting the typo. We have changed “TFN” to “TNF” in line 120.

After page 5, there are no line numbers.

Response: We have added line numbers after page 5.

The authors mentioned “NSAIDs are generally not considered viable option inf the long-term treatment of DED and should not be prescribed without caution and close monitoring [3,108,112].” What does the phrase, “viable option inf the long term treatment” mean? Please explain the meaning of a word, “inf”.

Response: We have updated “inf” to “for” in line 282.

Reference No.113 The description of the reference No.113 is incomplete. Example: FDA, FDA, ----- sicca (., etc.

Response: We have updated reference No. 115 (previously 113).

Reference No.141. The description of the reference No.141 is incomplete. (Author’s name)

Response: We have updated reference No. 143 (previously 141).